## Replications

psychology

deliberate practice, purposeful practice, expertise, replication, music, talent identification

**Author for correspondence:**
Brooke N. Macnamara
e-mail: brooke.macnamara@case.edu

# The role of deliberate practice in expert performance: revisiting Ericsson, Krampe & Tesch-Römer (1993)

## Brooke N. Macnamara and Megha Maitra

Department of Psychological Sciences, Case Western Reserve University, 10900 Euclid Avenue, Cleveland, OH 44106-7123, USA

BNM, 0000-0003-1056-4996

We sought to replicate Ericsson, Krampe & Tesch-Römer's (Ericsson, Krampe & Tesch-Römer 1993 *Psychol. Rev.* **100**, 363–406) seminal study on deliberate practice. Ericsson *et al.* found that differences in retrospective estimates of accumulated amounts of deliberate practice corresponded to each skill level of student violinists. They concluded, 'individual differences in ultimate performance can largely be accounted for by differential amounts of past and current levels of practice' (p. 392). We reproduced the methodology with notable exceptions, namely (i) employing a double-blind procedure, (ii) conducting analyses better suited to the study design, and (iii) testing previously unanswered questions about teacher-designed practice—that is, we examined the way Ericsson *et al.* operationalized deliberate practice (practice alone), and their theoretical but previously unmeasured definition of deliberate practice (teacher-designed practice), and compared them. We did not replicate the core finding that accumulated amounts of deliberate practice corresponded to each skill level. Overall, the size of the effect was substantial, but considerably smaller than the original study's effect size. Teacher-designed practice was perceived as less relevant to improving performance on the violin than practice alone. Further, amount of teacher-designed practice did not account for more variance in performance than amount of practice alone. Implications for the deliberate practice theory are discussed.

## 1. Introduction

The question of how people acquire expertise in complex domains such as music, sports and science has long been of interest to psychologists. A quarter century ago, in their classic article, Ericsson, Krampe and Tesch-Römer [1] introduced the highly

influential *deliberate practice view* in an attempt to answer this question. They posited that '[i]ndividual differences, even among elite performers, are closely related to assessed amounts of deliberate practice'—activities designed to improve performance (p. 363). Indeed, making exceptions only for height and body size for some sports, they wrote, 'we reject any important role for innate ability' (p. 399) and concluded, '[o]ur theoretical framework can also provide a *sufficient account* of the major facts about the nature and scarcity of exceptional performance' (p. 392, emphasis added).

The impact of this article—which shifted the narrative about the origins of expertise away from any important role for genes or stable abilities and towards the importance of practice and training—is difficult to overstate. Cited over 9000 times (source: Google Scholar as of 12 November 2018), it is one of the most referenced articles in the psychological literature. Moreover, the deliberate practice view gained substantial attention outside of the academic literature, inspiring numerous popular books including Geoff Colvin's *Talent Is Overrated* [2] and Malcolm Gladwell's *Outliers* [3], where Gladwell described the now famous '10 000 hour rule', i.e. with 10 000 hours of deliberate practice, one becomes an expert. It seems fair to say that no single article has had a greater impact on scientific and popular views of expertise than Ericsson *et al.* [1].

## 1.1. The original study

The deliberate practice theory proposes that expert performance is the result of prolonged practice, and that differences in levels of expertise are the result of differences in amounts of practice. The claim that some form of experience is necessary for an individual's improvement in a domain (i.e. *intra*-individual change) is indisputable. However, the claim that practice largely accounts for differences in performance across people (i.e. *inter*-individual differences) even at elite levels of performance is controversial (e.g. [4–6]).

In the first study to test the deliberate practice theory, Ericsson *et al.* [1] (Study 1) recruited three groups of 10 violin students from the Music Academy of West Berlin. The students in all three groups were described as 'expert violinists' (p. 373, 374), but were grouped based on skill level: the most elite 'best' group, the 'good' group and the least accomplished future 'music teacher' group. Ericsson *et al.* asked the violinists to describe their biographical histories, rate a number of activities, keep a diary for a week and estimate amount of time spent on current activities. Most importantly, they asked the violinists to retrospectively estimate weekly amounts of practice alone since taking up the violin. They multiplied these estimates by weeks in a year and summed across years for accumulated practice estimates. Ericsson *et al.* [1] concluded, 'there is complete correspondence between the skill level of the groups and their average accumulation of practice time alone with the violin' (p. 379).

## 1.2. Present study

### 1.2.1. Magnitude of effects

The present study is motivated by several factors. First, at least in retrospect, Ericsson *et al.*'s [1] finding that accumulated amount of deliberate practice differentiated even *experts* is surprising. That is, finding differences in amounts of practice between novices and experts would be expected, but finding significant differences among three groups of experts of varying accomplishment is less expected. For example, in a recent meta-analysis, Macnamara *et al.* [7] found that accumulated amount of deliberate practice significantly accounted for performance variance among sub-elite athletes (i.e. club-, local- and state/provincial-level athletes) and athletes with a range of skill levels, but did not reliably differentiate among elite athletes (i.e. national-, international- and Olympic/world champion-level athletes).

The magnitude of the effect of deliberate practice reported by Ericsson *et al.* [1] is also surprising. Another meta-analysis [6] found that the average correlation between accumulated deliberate practice and performance in music was 0.48, 95% CI [0.38, 0.56]. The calculated effect size from Ericsson *et al.*'s study of violinists was 0.70, substantially higher than the 95% confidence interval's upper limit found by the meta-analysis. We thus sought to replicate Ericsson *et al.*'s finding, because when surprising findings are replicated, confidence in an existing theory increases; however, when surprising findings are not replicated, this can spur development of improved theories [8].

### 1.2.2. Potential bias

Ericsson *et al.*'s [1] method for collecting retrospective estimates of practice—a structured interview—is potentially prone to experimenter bias and response bias [9–11]. That is, experimenters aware of the

**Table 1.** Example inconsistent definitions of 'deliberate practice'.

| quote | reference |
| --- | --- |
| practice activities require a teacher | |
| 'Ericsson et al. [1993] defined *deliberate practice* as a very specific activity designed for an individual by a skilled teacher explicitly to improve performance' | Krampe & Ericsson [13, p. 333] |
| 'Ericsson et al. (1993) identified activities that met the necessary requirements for effective training and were designed by a teacher to improve a specific individual's performance. They termed these activities "deliberate practice"' | Ericsson [15, p. 368] |
| 'When this type of training is supervised and guided by a teacher, it is called "deliberate practice"—a concept my colleagues and I introduced in 1993' | Ericsson [16, p. 1472] |
| practice activities do *not* require a teacher | |
| 'Ericsson et al. (1993) proposed the term deliberate practice to refer to those training activities that were designed solely for the purpose of improving individuals' performance by a teacher or the performers themselves' | Ericsson [14, p. 84] |
| 'Ericsson et al. (1993) introduced the term deliberate practice to describe focused and effortful practice activities that are pursued with the explicit goal of performance improvement. Deliberate practice implies that well-defined tasks are practised at an appropriate level of difficulty and that informative feedback is given to monitor improvement. These activities can be designed by external agents, such as teachers or trainers, or by the performers themselves' | Keith & Ericsson [17, p. 136] |
| 'it has been possible to identify special practice activities (deliberate practice) that performers' teachers or the performers themselves design' | Ericsson [18, p. 1128] |

hypothesis that accumulated practice will correspond to skill level and aware of the violinists' skill group can unconsciously influence participants' estimates in an interview procedure. There is no indication in Ericsson *et al.* that experimenters were blind to the participants' skill level. Experimenters in Ericsson *et al.* also provided participants with a '[d]escription of the institute and the purpose of the study' just before the interview began [12, p. 151]. Depending on what was said, this could have influenced participants' estimates. To reduce potential experimenter-expectancy bias and response bias, the present study employed a double-blind procedure—experimenters were unaware of violinists' skill groups and violinists were not told the purpose of the study or that there were multiple skill groups.

### 1.2.3. Multiple definitions

Ericsson *et al.* [1] appear to theoretically define deliberate practice as practice activities designed by a teacher. For example, they state, 'the teacher designs practice activities that the individual can engage in between meetings with the teacher. We call these practice activities *deliberate practice*' (p. 368). However, according to both the study's Methods section and the interview protocol for Ericsson *et al.* [1] that can be found in Krampe's dissertation's appendix [12], Ericsson *et al.* [1] appear to operationally define deliberate practice as 'practice alone' with no indication that participants were asked to restrict their estimates of practice to only those designed by a teacher. Participants' estimates of amounts of 'practice alone' are the key outcome variable for all analyses in support of deliberate practice.

Examining subsequent literature by Ericsson and colleagues to determine the single definition of deliberate practice does not clarify whether (i) practice activities *need to* be designed by a teacher to qualify as deliberate practice, or (ii) practice activities *need not* be designed by a teacher. For example, Krampe & Ericsson [13] stated, 'Ericsson et al. [1993] defined *deliberate practice* as a very specific activity designed for an individual by a skilled teacher' (p. 333). By contrast, Ericsson [14] stated, 'Ericsson et al. (1993) proposed the term deliberate practice to refer to those training activities that were designed solely for the purpose of improving individuals' performance by a teacher *or the performers themselves*' (p. 84, emphasis added). See table 1 for more examples.

In the present study, we first asked participants to estimate amounts of deliberate practice defined as *practice alone* with no restrictions that the activities be teacher designed, replicating Ericsson *et al.*'s [1] reported methods. We next asked participants to estimate amounts of deliberate practice defined as *teacher-designed practice*, restricting estimates to time spent on practice activities that had been designed by a teacher. Furthermore, we sought to measure the correlation between estimates of practice alone and estimates of teacher-designed practice and to test the extent to which each estimate explained variance in performance. While this additional measure was a departure from the original methods, it seems necessary to establish whether these measures tap the same or different constructs, as both measures are described in subsequent accounts of the original study (table 1).

### 1.2.4. Issues with statistical analyses

Ericsson *et al.* performed two separate *F*-tests on variables of interest. In the first *F*-test, they compared the 10 best violinists to the 10 good violinists, excluding the least-accomplished group, but using the full sample degrees of freedom for the denominator. (This technique slightly reduces the critical *F*-value needed to reach significance.) In the second *F*-test, Ericsson *et al.* combined the 10 best violinists and the 10 good violinists into a single group of 20 violinists, then compared this combined group with the 10 least-accomplished violinists. Two separate *F*-tests make it difficult to determine if there is a main effect of group or whether the good violinists' scores are significantly higher than the less accomplished violinists' scores. We therefore conducted analyses that simultaneously included all three groups as well as planned comparisons to test whether the best violinists practised significantly more than the good violinists, and whether the good violinists in turn practised significantly more than the less accomplished violinists, as is concluded in Ericsson *et al.* [1].

## 2. Methods

This study has two preregistrations. All hypotheses, methods and planned analyses were preregistered in detail on the Open Science Framework prior to viewing the data. This preregistration is available at https://osf.io/khjs7. Following data analysis, this article received results-blind in-principle acceptance at Royal Society Open Science. Following the in-principle acceptance, the accepted Stage 1 version of the manuscript, not including results and discussion, was preregistered on the Open Science Framework following Royal Society Open Science's Replication article type, Results-Blind track protocol. This preregistration is available at https://osf.io/jyn5w. Materials used are openly available at https://osf.io/4595q. This study was approved by Case Western Reserve University's Institutional Review Board.

### 2.1. Participants

#### 2.1.1. The best violinists

To recruit participants, Ericsson *et al.* [1] contacted the Music Academy of West Berlin, an academy whose violin-training programme boasted an international reputation. Ericsson *et al.* [1] asked faculty from the school to nominate violin students 'who had the potential for careers as international soloists' (p. 373). Of the 14 students nominated, 10 agreed to participate. Ericsson *et al.* [1] referred to this group of students as the 'best violinists'.

Following Ericsson *et al.*'s [1] methodology as closely as possible, we contacted the Cleveland Institute of Music, a highly ranked music conservatory, whose violin-training programme is internationally acclaimed (e.g. [19]). We asked faculty from the school to nominate violin students who had the potential for careers as international soloists. Of the 24 students nominated, 13 agreed to participate. We follow Ericsson *et al.*'s terminology and refer to this group of violin students as the 'best violinists'.

#### 2.1.2. The good violinists

The professors at the Music Academy of West Berlin also nominated a large number of good student violinists from the same department. Ericsson *et al.* [1] selected 10 students from this group, matching sex and age to the 10 'best violinists'. Ericsson *et al.* referred to this group of students as the 'good violinists'.

Following Ericsson *et al.*'s [1] methodology, we asked the professors to also nominate good student violinists from the same department. We selected 13 students from this group, matching sex and age as much as possible to our 13 'best violinists'. We follow Ericsson *et al.*'s terminology and refer to this group of violin students as the 'good violinists'.

### 2.1.3. The less accomplished violinists

Ericsson *et al.* [1] recruited 10 student violinists, matching sex and age with the other two groups, from a different department at the school (music education). This department had lower music performance admission standards than the department from which the best and good violinists were recruited. Ericsson *et al.* referred to the violinists in this least accomplished group as the 'the music teachers' because teaching was the most likely future career of these students.

The Cleveland Institute of Music does not have a music education department and so we recruited violin students from the neighbouring music department at Case Western Reserve University. Like the music education department from Ericsson *et al.* [1], the music department at Case Western Reserve University has lower music performance admission standards than the Cleveland Institute of Music from which the best and good violinists were recruited (see, e.g. www.best-music-colleges.com/case-western-reserve-university). The Cleveland Institute of Music and Case Western Reserve University are affiliated institutions with a joint music programme and many of the music professors are associated with both departments at the two schools. Additionally, many Cleveland Institute of Music students are also Case Western Reserve University students (though their music department memberships differ). Of the 14 student violinists from Case Western Reserve University's music department, 13 agreed to participate. Not all these violin students were pursuing music education (many were pursuing music performance careers like the other two groups), and thus, we call this group the less accomplished violinists.

Table 2 compares the samples from the original study and the current study.[1]

### 2.1.4. Sample size

The current sample was obtained from over 4 years of recruiting efforts (summer of 2014–autumn of 2018). While our sample size is 33% larger than Ericsson *et al.*'s, it is still small. Conducting expertise research, by definition, means studying a small subset of the population, thus there are relatively few participants from which to sample, making large-sample replications nearly impossible. However, if replications of expertise studies never enter the scientific record because of small samples, then we probably will never provide additional evidence to support or refute the original studies. This leaves only the original study (with an even smaller sample size) as the only indicator in the scientific record.

To be clear, evidence from small samples—in original studies or replications—should be interpreted with caution. However, when the original study (with a small sample) has already entered the scientific record, replications with similar sample sizes (since that is all that is feasible) should also be allowed to enter the scientific record rather than be suppressed from publication. This is probably the best way to accumulate knowledge in this area.

Further, the original publication made clear and bold claims, such as: 'it is impossible for an individual with less accumulated practice at some age to catch up with the best individuals, who have started earlier and maintain maximal levels of deliberate practice not leading to exhaustion' (p. 393). Thus, while a similar finding from a small replication study would add minimal support, evidence contradicting an 'impossibility' needs only a single example to falsify it. In this way, replications of any size have the potential to contribute to our understanding in this area.

Our sample size is large enough to detect an effect size of $\eta^2 = 0.48$, which is the effect size found by Ericsson *et al.* [1], with greater than 99.9% power. Indeed, based on Ericsson *et al.*'s claims, such as '[i]ndividual differences, even among elite performers, are closely related to assessed amounts of deliberate practice', this suggests that these large effects should be easily detectable. Further, our sample size is large enough to detect an effect size of $\eta^2 = 0.23$, which is the average effect size for deliberate practice on performance in the domain of music [6], with 82% power.

---

[1]Ericsson *et al.* [1] also recruited 10 professional violinists. However, they were not included in any inferential analyses except for reporting there were no differences from the student violinists on some demographic variables and reporting there was no difference between them and the best violinists on accumulated practice alone until age 18.

**Table 2.** Characteristics of the three groups of violinists in Ericsson *et al*. [1] and present study. Note: 95% CI, 95% confidence interval lower and upper bounds.

| | Ericsson *et al*. [1] | present study |
|---|---|---|
| **best violinists** | | |
| description | faculty-nominated violin students at the Music Academy of West Berlin who had the potential for careers as international soloists | faculty-nominated violin students at the Cleveland Institute of Music who had the potential for careers as international soloists |
| sample size | $n = 10$ <br> 7 females, 3 males | $n = 13$ <br> 6 females, 7 males |
| age | $M$ = not reported; 95% CI = not reported; range = not reported | $M = 21.85$; 95% CI [21.02, 22.67]; range = 20–25 |
| **good violinists** | | |
| description | faculty-nominated good violinists from the same department as the best violinists | faculty-nominated good violinists from the same department as the best violinists |
| sample size | $n = 10$ <br> 7 females, 3 males | $n = 13$ <br> 6 females, 7 males |
| age | $M$ = not reported; 95% CI = not reported; range = not reported | $M = 20.54$; 95% CI [19.31, 21.77]; range = 18–25 |
| **less accomplished violinists** | | |
| description | students from the department of music education from the same institution | students from the department of music at Case Western Reserve University, an affiliated institution |
| sample size | $n = 10$ <br> 7 females, 3 males | $n = 13$ <br> 8 females, 5 males |
| age | $M$ = not reported; 95% CI = not reported; range = not reported | $M = 20.00$; 95% CI [18.85, 21.15]; range = 18–26 |
| **total sample** | | |
| sample size | $N = 30$ | $N = 39$ |
| age | $M = 23.1$; 95% CI = not reported; range = 18–not reported; 'no reliable differences in age' among the three groups (p. 374); statistics not reported | $M = 20.79$; 95% CI [20.14, 21.45]; range = 18–26 no reliable differences in age among the three groups; $F_{2,36} = 2.95$, $p = 0.065$ |

## 2.2. Materials and procedure

Following the procedure of Ericsson *et al*. [1] described in their Methods section, in the first session of our study, we asked participants to briefly describe their musical histories. Next, specific biographical information was collected via a structured interview, including the age they began playing the violin and any other musical instrument, ages they changed violin instructors and experience participating in music competitions. Participants were then asked to estimate the number of hours per week they had practised alone with the violin for each year beginning with the age they first started practising to the present. This is the key measure, after multiplying by weeks in a year and summing across years, used in the original study to make claims about the importance of accumulated deliberate practice.

Differing from Ericsson *et al*. [1], after participants gave their complete retrospective estimates of practice alone with the violin, we asked participants to estimate the number of hours per week they engaged in teacher-designed practice with the violin for each year beginning with the age they first began practising to the present. Estimates of teacher-designed practice alone were elicited *after* estimates of practice alone, so that the original measure would not be influenced by this additional

measure. We also obscured the column title 'Practice Designed by Teacher' the researcher used to record information while participants gave their practice alone estimates, so that this would not influence their practice alone estimates.

Teacher-designed practice was described as practice activities designed by an instructor and was differentiated from self-guided practice. The interviewer explained that estimates of practice alone and estimates of teacher-designed practice could overlap to any degree, that is, amounts of teacher-designed practice could be less than amounts of practice alone (i.e. if some time spent practising alone was self-guided rather than teacher-designed), the same as amounts of practice alone (i.e. if all time spent practising alone consisted of practising teacher-designed activities), or greater than amounts of practice alone (i.e. if time spent on teacher-designed activities was conducted both alone and with others). Interview materials are openly available at osf.io/4595q.

As in Ericsson *et al.*, we asked participants to estimate current typical weekly time spent on a variety of musical and everyday activities as well as to rate the activities on a scale of 0–10 in terms of relevance to improving performance on the violin, the effort to perform the activity and enjoyableness of performing the activity. The first rating dimension was the relevance of the activity for improving performance on the violin. The second rating dimension was the effort required to perform the activity. The third rating dimension was the enjoyableness of the activity without considering the outcome of the activity. We provided the same example as Ericsson *et al.* to describe this: '… it is possible to enjoy the result of having cleaned one's house without enjoying the activity of cleaning' (p. 373). Participants playing other instruments besides the violin provided estimates of hours and ratings first for activities involving the violin and then for activities involving all other instruments. Differing from Ericsson *et al.*, we also asked for current estimates and ratings of time typically spent engaged in teacher-designed practice activities after asking about practice alone. We also slightly altered and clarified some of the everyday activity categories unrelated to practice such as adding a category for social activities and a category for social media and email (which were not prevalent in 1993). See table 3.

As in Ericson *et al.* [1], during the second session, participants answered questions about practice and concentration, the number of minutes of violin music they could play from memory and recalled their activities (table 3) in the previous 24 h period using a diary sheet divided into 96 15 min intervals. Participants were then given instruction on maintaining diary logs for the next 7 days, including coding each activity according to the set of musical and everyday activities previously rated.

Ericsson *et al.* [1] designed diary sheets where each sheet represented a 24 h day, divided into 96 15 min intervals. Participants were instructed to fill in the sheets with their activities and were given seven envelopes addressed to the experimenters to be mailed back each day. Participants worked with copies of their diary sheets and coded the activities based on the taxonomy of activities (table 3, left column) before the third and final session. We designed diary sheets using Excel, where each sheet represented a 24 h day, divided into 96 15 min intervals. The experimenter emailed a diary sheet to the participant each day for the following day and instructed the participants to email back the completed sheets each day. After receiving the seven diary logs, the experimenter emailed back the seven sheets in one document (one tab per day) along with the taxonomy of activities (table 3, right column) and asked participants to code each activity based on the taxonomy. We encouraged participants to identify the primary, or most relevant, category for each activity but allowed them to use more than one when appropriate (e.g. discussing music theory over lunch). Our diary log procedure was identical to Ericsson *et al.*'s except that we used Excel sheets and email rather than paper sheets, copies, envelopes, stamps and postal mail.

During the third session, Ericsson *et al.* allowed participants to ask any questions they had about their activities coding and then they asked participants about developmental life goals. We allowed participants to ask any questions they had about their activities coding and then asked participants about developmental life goals. Following all other replication measures, we administered several other measures (see electronic supplementary material). We then engaged them in a general debriefing.

## 3. Results

Open data are available at osf.io/4595q. Additional results can be found in the electronic supplementary material. In cases where assumptions are not met, we use non-parametric tests: Kruskal–Wallis tests in place of between-subject ANOVAs, Greenhouse–Geisser corrections for repeated-measures ANOVAs and Welch's tests in place of Student's *t*-tests.

**Table 3.** Taxonomy of activities.

| Ericsson *et al.* [1] | present study |
| --- | --- |
| music related | music related |
| practice (alone) | practice (alone) |
| — | practice (activities designed by teacher) |
| practice (with others) | practice (with others) |
| playing for fun (alone) | playing for fun (alone) |
| playing for fun (with others) | playing for fun (with others) |
| taking lessons | taking lessons |
| giving lessons | giving lessons |
| solo performance | solo performance |
| group performance | group performance |
| listening to music | listening to music |
| music theory | music theory |
| professional conversation | professional conversation |
| organization and preparation | organization and preparation |
| everyday | everyday |
| household chores | household chores |
| child care | child care |
| shopping | shopping |
| work (not music related) | work (not music related) |
| sports | sports/fitness |
| body care and health | personal care |
| sleep | sleep |
| education (not music) | education (not music) |
| committee work | committee work |
| — | social activities |
| — | social media/email |
| leisure | leisure/hobbies |

## 3.1. General music histories

### 3.1.1. Music demographics

*Original study analyses*. For the following variables—age began practising the violin; age began violin lessons; age decided to pursue music as a career; number of violin teachers; number of other instruments played; and years practising the violin—Ericsson *et al.* [1] report the grand means and that there were 'no systematic differences between groups' (p. 374). Ericsson does not report group-level descriptive statistics.

*Replication study analyses*. For the following variables—age began practising the violin; age began violin lessons; age decided to pursue music as a career; number of violin teachers; number of other instruments played; and years practising the violin—we report grand means, 95% confidence intervals, ranges and test statistics for group differences in table 4. Group-level descriptive statistics are also reported in table 4.

While not enough information was given in Ericsson *et al.* [1] to statistically compare their student violinists to the present study's student violinists, the samples appear similar. Numerically, our violinists were slightly younger and began playing the violin slightly earlier.

**Table 4.** Comparison of musical backgrounds between Ericsson et al. [1] and the present study and among the skill groups in the present study. Note: 95% CI, 95% confidence interval. Numbers in brackets represent the lower and upper bounds of the confidence interval. Ericsson et al. [1] do not report any variance statistics (e.g. standard deviations, 95% confidence intervals) for the variables listed here nor any test statistics for group differences. For 'all violinists', Ericsson et al. [1] report the grand means for the three student groups and a group of professional violinists combined; ours includes only the student violinists. See electronic supplementary material for additional results. Not enough information on the whole sample statistics is reported in Ericsson et al. [1] to test whether there are any significant differences between our samples' musical histories and Ericsson et al.'s.

| | all violinists | | present study | | |
| | Ericsson et al. [1] | | | | |
| | M | group comparison | M | 95% CI range group comparison | range |
|---|---|---|---|---|---|
| age began practising the violin | 7.9 | 'no systematic differences between groups' (p. 374) | 5.51 | [4.74, 6.28] $F_{2,36} = 0.11$, $p = 0.893$, $\eta^2 = 0.01$ | 2–11 |
| age began violin lessons | 8.0 | 'no systematic differences between groups' (p. 374) | 5.64 | [4.86, 6.42] $F_{2,36} = 0.25$, $p = 0.784$, $\eta^2 = 0.01$ | 2–11 |
| age decided to pursue music as a career[a] | 14.9 | 'no systematic differences between groups' (p. 374) | 13.64 | [12.30, 14.98] $F_{2,33} = 2.73$, $p = 0.080$, $\eta^2 = 0.14$ | 2–20 |
| number of violin teachers | 4.1 | 'no systematic differences between groups' (p. 374) | 5.08 | [4.48, 5.68] $F_{2,36} = 2.17$, $p = 0.129$, $\eta^2 = 0.11$ | 3–12 |
| number of other instruments played | 1.8 | 'no systematic differences between groups' (p. 374) | 1.05 | [0.68, 1.42] $\chi^2_2 = 3.24$, $p = 0.198$, $\eta^2 = 0.11$ | 0–6 |
| years practising the violin | by the mean age of 23, all had ≥10 years violin practice | | by the mean age of 21, all had ≥10 years violin practice | | |

(Continued.)

**Table 4.** (*Continued.*)

| | present study | | | | | | | | |
| | best violinists | | | good violinists | | | less accomplished | | |
| | M | 95% CI | range | M | 95% CI | range | M | 95% CI | range |
|---|---|---|---|---|---|---|---|---|---|
| age began practising the violin | 5.46 | [3.93, 6.99] | 2–11 | 5.31 | [3.95, 6.66] | 2–10 | 5.77 | [4.57, 6.97] | 3–10 |
| age began violin lessons | 5.62 | [4.01, 7.22] | 2–11 | 5.31 | [3.95, 6.66] | 2–10 | 6.00 | [4.89, 7.11] | 4–10 |
| age decided to pursue music as a career[a] | 12.77 | [10.14, 15.40] | 2–10 | 12.62 | [10.52, 14.72] | 8–16 | 16.10 | [14.66, 17.54] | 12–19 |
| number of violin teachers | 5.69 | [4.67, 6.72] | 3–10 | 5.31 | [4.08, 6.53] | 3–12 | 4.23 | [3.52, 4.94] | 3–6 |
| number of other instruments played | 0.62 | [0.26, 0.97] | 0–2 | 1.00 | [0.62, 1.38] | 0–2 | 1.54 | [0.61, 2.47] | 0–6 |
| years practising the violin | 16.38 | [14.79, 17.98] | 11–20 | 15.23 | [14.03, 16.43] | 11–18 | 14.23 | [12.55, 15.91] | 10–20 |

[a]Three violinists (all less accomplished violinists) had not (yet) decided to pursue a career in music. One of the best violinists said she had always wanted to pursue a career in music, so we used the age she began practising the violin as the age she decided to pursue a career in music.

### 3.1.2. Competitions

*Original study analyses*. Ericsson *et al.* reported the means of the number of competitions successfully entered for each of the three student groups. They then conducted two ANOVAs. The first ANOVA tested whether there was a difference in the number of music competitions entered between the best violinists and the good violinists. This analysis excluded the less accomplished violinists but used the full sample degrees of freedom. For the second ANOVA, the best and good violinists were combined into a single group. The ANOVA tested whether there was a difference in the number of music competitions entered between the combined best-and-good-violinists group and the less accomplished violinists. Based on Ericsson *et al.*'s reported means (best = 2.9, good = 0.6, less accomplished = 0.2), many violinists reported 0 competition entries and so it is unlikely that this variable followed a normal distribution for each group. Therefore, ANOVAs were probably not appropriate for Ericsson *et al.*'s [1] data for this measure.

*Replication study analyses*. The student violinists in the present study had successfully entered numerically more violin competitions ($M = 8.41$, 95% CI [5.97, 10.85], range = 0–35) than the student violinists in Ericsson *et al.* [1] ($M = 1.23$). These numbers followed normal distributions for each group (all skews and kurtoses less than |1.25|). The best violinists averaged 13.31 (95% CI [8.25, 18.37], range = 1–35) successful entries, followed by the good violinists who averaged 8.69 (95% CI [5.79, 11.60], range = 0–20), followed by the less accomplished violinists who averaged 3.23 (95% CI [0.73, 5.73], range = 0–12). The difference across groups was significant, $F_{2,36} = 7.27$, $p = 0.002$, $\eta^2 = 0.29$. Planned comparisons revealed that while the best violinists had competed in more competitions than the good violinists on average, this difference was not significant, $t_{24} = 1.55$, $p = 0.134$, $d = 0.63$. The good violinists had competed in significantly more competitions than the less accomplished violinists on average, $t_{24} = 2.79$, $p = 0.010$, $d = 1.14$.

## 3.2. Activity ratings

*Original study analyses*. Ericsson *et al.* tested whether there was a significant interaction between student group and activity in their ratings. They did not find significant interactions (statistical results not reported) and so then collapsed across the three student groups' ratings for each activity. For each rating type (relevance, effort, enjoyment), Ericsson *et al.* tested whether the mean activity rating was significantly higher or lower than the grand mean activity rating, adjusting α for multiple comparisons. They reported means in a table.

*Replication study analyses*. We tested whether there was a significant interaction between student group and activity in their ratings. We found no significant interactions. See electronic supplementary material for activity rating by group interaction test statistics. As with Ericsson *et al.* [1], we collapse across the three student groups' ratings for each activity. Table 5 compares the mean ratings with Ericsson *et al.*'s mean ratings and indicates whether the mean activity rating was significantly higher or lower than the grand mean activity rating, adjusting $\alpha$ for multiple comparisons.

As can be seen in table 5, Ericsson *et al.*'s sample ratings and the present sample ratings are largely similar, especially on key activities. As with Ericsson *et al.* [1], participants in the current study rated practice alone as the most relevant to improving performance on the violin. The average relevance rating for practice alone from the current study was 9.87 (95% CI [9.67, 10.00²], range = 7–10). Likewise, in both studies, taking lessons was rated as the second-most relevant activity to improving performance on the violin (present study $M = 9.79$, 95% CI [9.71, 9.88], range = 8–10). In both studies, solo performance was rated as the third-most relevant activity for improving performance on the violin (present study $M = 9.41$, 95% CI [8.95, 9.87], range = 3–10). The musical activity added to the present study—teacher-designed practice—had the fourth-highest average relevance rating ($M = 9.38$, 95% CI [8.99, 9.77], range = 5–10), and was also rated significantly higher than the grand mean.

## 3.3. Current estimates of practice

### 3.3.1. Weekly estimates of practice alone

*Original study analyses*. Ericsson *et al.* reported that there was no significant difference between the best violinists ($M$ not reported) and good violinists ($M$ not reported) (statistical test not reported).

²The upper bound for the 95% confidence interval was calculated at 10.05. However, we report this as 10.00 because 10 was the scale's maximum rating.

**Table 5.** Comparison of activity ratings between Ericsson *et al.* [1] and the present study. Note: 1993, Ericsson *et al.*'s [1] rating results; present, the present study's rating results; H, significantly higher than the grand mean; L, significantly lower than the grand mean.

| | relevance | | effort | | enjoyment | |
|---|---|---|---|---|---|---|
| | 1993 | present | 1993 | present | 1993 | present |
| **musical activities** | | | | | | |
| practice alone | 9.82 H | 9.87 H | 8.00 H | 8.29 H | 7.23 | 7.27 |
| teacher-designed practice | N/A | 9.38 H | N/A | 8.21 H | N/A | 6.49 |
| practice with others | 8.73 H | 8.62 H | 6.97 H | 7.10 H | 7.57 | 7.87 |
| playing for fun alone | 5.67 | 6.65 | 3.27 L | 3.44 L | 8.33 H | 8.54 H |
| playing for fun with others | 6.67 | 6.38 | 3.93 | 4.23 L | 8.60 H | 8.86 H |
| taking lessons | 9.63 H | 9.79 H | 8.60 H | 8.68 H | 7.67 | 8.00 H |
| giving lessons | 7.03 | 7.90 H | 7.51 H | 8.36 H | 6.79 | 6.64 |
| solo performance | 9.03 H | 9.41 H | 9.80 H | 9.69 H | 7.28 | 7.85 |
| group performance | 7.67 H | 8.67 H | 8.14 H | 8.09 H | 8.07 H | 8.36 H |
| listening to music | 8.33 H | 8.33 H | 4.38 | 2.92 L | 8.38 H | 8.92 H |
| music theory | 7.63 H | 7.44 H | 6.37 H | 7.85 H | 6.07 | 5.23 L |
| professional conversation | 6.50 | 6.26 | 4.33 | 4.88 | 6.40 | 6.77 |
| organization + prep | 2.90 L | 7.31 | 4.70 | 6.33 | 1.53 L | 5.56 L |
| **everyday activities** | | | | | | |
| household chores | 1.80 L | 2.79 L | 2.23 L | 6.18 | 3.63 L | 4.23 L |
| child care | 2.64 L | 1.82 L | 6.14 | 9.15 H | 6.43 | 5.95 |
| shopping | 0.77 L | 2.41 L | 2.80 L | 3.51 L | 3.97 L | 6.92 |
| work (not music related) | 1.79 L | 1.87 L | 5.56 | 6.36 | 3.74 L | 5.04 L |
| sports/fitness[a] | 6.07 | 5.74 | 2.67 L | 7.72 H | 7.07 | 6.56 |
| personal care[a] | 4.90 | 6.18 | 1.43 L | 4.68 | 5.23 | 6.47 |
| Sleep | 8.17 H | 8.49 H | 0.47 L | 2.33 L | 7.70 | 9.23 H |
| education (not music) | 4.52 | 4.29 L | 5.45 | 7.53 H | 7.17 | 5.72 L |
| committee work | 1.93 L | 2.18 L | 5.55 | 6.44 | 5.07 | 5.38 L |
| social activities | N/A | 4.03 L | N/A | 3.54 L | N/A | 8.44 H |
| social media/email | N/A | 3.67 L | N/A | 3.05 L | N/A | 6.21 |
| leisure/hobbies[a] | 6.30 | 4.23 L | 3.00 L | 2.64 L | 8.93 H | 9.08 H |
| grand mean | 5.89 | 6.15 | 5.03 | 6.05 | 6.52 | 7.02 |

[a]This category is slightly altered from Ericsson *et al.*'s (see table 3 for comparison).

They report the results of the second ANOVA combining the best and good violinists into a single group (29.8 h) and comparing their combined mean weekly hours to the less accomplished violinists' mean weekly hours of practice alone (13.4 h), which was significant at the $p < 0.001$ level.

*Replication study analyses*. We found a significant effect of group in estimates of typical weekly practice alone, $F_{2,36} = 27.36$, $p < 0.001$, $\eta^2 = 0.60$. Unlike Ericsson *et al.* [1], we found a significant difference between the best violinists ($M = 22.69$, 95% CI [17.65, 27.74], range = 8.00–40.00) and the good violinists ($M = 30.65$, 95% CI [27.04, 34.27], range = 20.00–45.50), $p = 0.024$, $d = -1.03$, such that the best violinists estimated spending significantly *less* time practising alone per week than the good violinists. Both the best and good violinists estimated significantly more practice time alone per week than the less accomplished violinists ($M = 9.58$, 95% CI [6.55, 12.61], range = 2.00–18.00), $p < 0.001$, $d = 1.78$, and $p < 0.001$, $d = 3.57$, respectively. See figure 1.

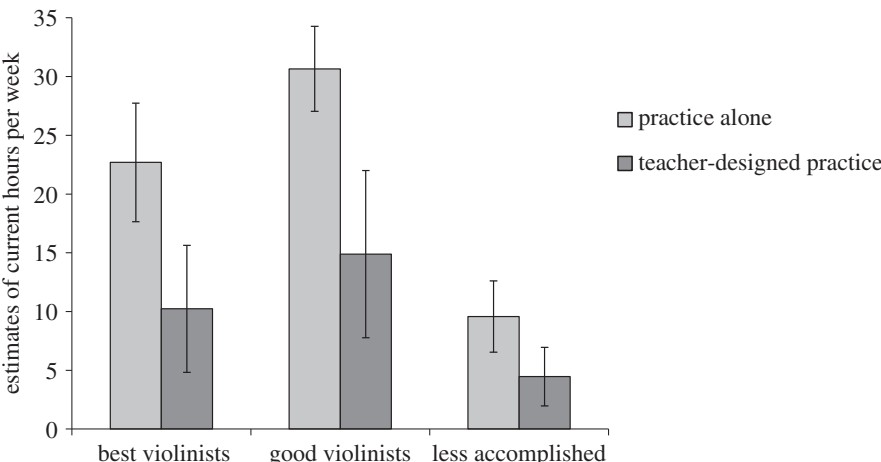

**Figure 1.** Estimates of current practice in a typical week. Error bars represent 95% confidence intervals.

### 3.3.2. Weekly estimates of teacher-designed practice

*Original study analyses.* Ericsson *et al.* [1] did not report estimates of teacher-designed practice independent of 'practice alone'.

*Replication study analyses.* When examining weekly estimates of teacher-designed practice, we did not observe a main effect of group, $\chi_2^2 = 3.14$, $p = 0.208$, $\eta^2 = 0.17$. The best violinists estimated 10.23 h of teacher-designed practice on average per week (95% CI [4.82, 15.64], range = 0.00–28.00), the good violinists estimated 14.88 h of teacher-designed practice per week (95% CI [7.77, 22.00], range = 0.00–35.00) and the less accomplished violinists estimated 4.46 h of teacher-designed practice on average per week (95% CI [1.96, 6.96], range = 0.00–15.00). See figure 1.

Estimates of typical amounts of weekly practice alone and estimates of typical amounts of weekly teacher-designed practice were moderately correlated, $r_{37} = 0.41$, $p = 0.009$. Participants estimated more time spent practising alone than engaging in teacher-designed practice, $F_{1,36} = 37.99$, $p < 0.001$, $\eta_p^2 = 0.51$. The difference in estimates (practice alone–teacher-designed practice) did not significantly interact with group, $F_{2,36} = 3.06$, $p = 0.060$, $\eta_p^2 = 0.15$.

## 3.4. Comparing weekly practice estimates with diary-recorded practice

*Original study analyses.* Ericsson *et al.* [1] report a significant repeated-measures ANOVA comparing weekly estimates and diary logs, indicating that the violinists estimated engaging in significantly more practice alone per week than they logged during the diary week at the $p < 0.001$ level, suggesting an overestimation bias. Importantly, they found that this overestimation bias did not differ across groups (statistical results not reported), suggesting that the estimates 'appear to be valid, albeit biased, indicators of actual practice' (p. 378).

*Replication study analyses.* Consistent with this finding, we observed that the estimates of current typical practice alone were significantly higher than amount of practice alone logged during the diary week, $F_{1,35} = 19.62$, $p < 0.001$, $\eta_p^2 = 0.36$, and that this overestimation did not differ across groups, $F_{2,35} = 1.45$, $p = 0.249$, $\eta_p^2 = 0.08$. See electronic supplementary material for additional diary activity-related results.

## 3.5. Retrospective estimates of practice during development

### 3.5.1. Weekly practice alone during development

*Original study analyses.* Ericsson *et al.* [1] illustrated retrospective estimates of weekly practice alone by age to age 20 and observed that practice alone increased monotonically.

*Replication study analyses.* Figure 2 illustrates the retrospective estimates of weekly practice alone by age to age 20. Like Ericsson *et al.* [1], practice alone generally increased monotonically.

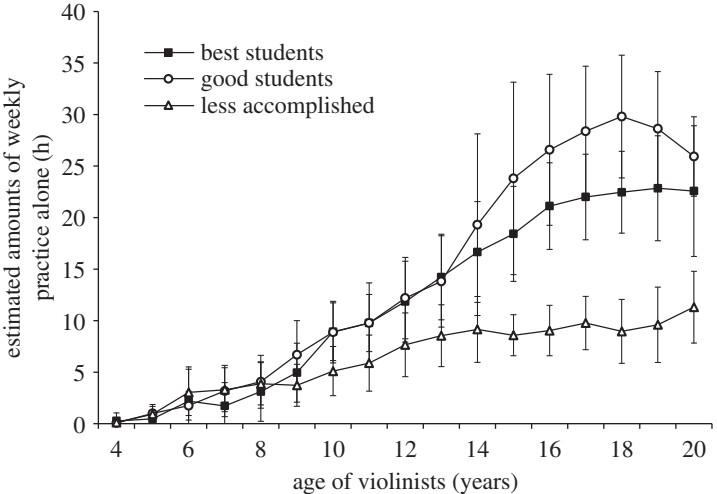

**Figure 2.** Estimated amounts of weekly practice alone with the violin as a function of age. Error bars represent 95% confidence intervals.

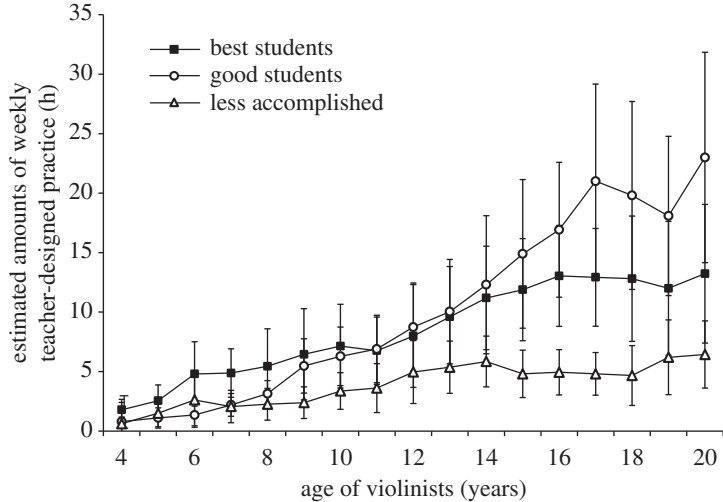

**Figure 3.** Estimated amounts of weekly teacher-designed practice with the violin as a function of age. Error bars represent 95% confidence intervals.

### 3.5.2. Weekly teacher-designed practice during development

*Original study analyses.* Ericsson *et al.* [1] did not report estimates of teacher-designed practice independent of 'practice alone'.

*Replication study analyses.* Figure 3 illustrates the retrospective estimates of weekly teacher-designed practice by age to age 20. Teacher-designed practice increased monotonically.

### 3.5.3. Accumulated practice alone

*Original study analyses.* The most important test of the deliberate practice theory is whether there are differences in *accumulated* practice between the skill groups [1]. Ericsson *et al.* estimated accumulated practice alone by multiplying weekly estimates by the number of weeks in a year and summing this amount to age 18. Age 18 was chosen to avoid any confounding influences of activities from the music academy.

They reported two ANOVAs. In the first ANOVA, they compared the best violinists' mean accumulated hours of practice alone ($M = 7410$) to the good violinists' mean accumulated hours of practice alone ($M = 5301$) (excluding the less accomplished violinists but using the full sample degrees of freedom), and found significance at the $p < 0.05$ level. In the second ANOVA, they combined the

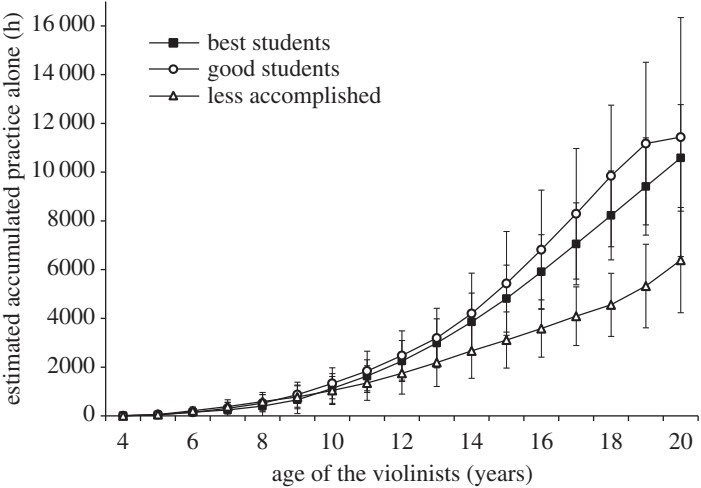

**Figure 4.** Accumulated amount of practice alone as a function of age for the three skill groups. Error bars represent 95% confidence intervals.

best and good violinists into a single group and compared their combined mean accumulated hours of practice alone to the less accomplished violinists' mean accumulated hours of practice alone ($M = 3420$), and found significance at the $p < 0.01$ level. They concluded, 'Hence, there is complete correspondence between the skill level of the groups and their average accumulation of practice time alone with the violin' (p. 379).

*Replication study analyses.* We found a significant effect of group for accumulated practice alone until age 18, $\chi_2^2 = 13.90$, $p = 0.001$, $\eta^2 = 0.26$. However, the best violinists ($M = 8224$, 95% CI [6400, 10 048], range = 3978–14 664) had *not* accumulated significantly more practice alone by age 18 than the good violinists ($M = 9844$, 95% CI [6937, 12 751], range = 3120–21 268), $t_{24} = -0.93$, $p = 0.364$, $d = -0.38$. The good violinists had accumulated significantly more practice alone by age 18 than the less accomplished violinists ($M = 4558$, 95% CI [3264, 5851], range = 2522–10 972), $t_{16.57} = 3.26$, $p = 0.005$, $d = 1.33$.

In figure 4, we plot accumulated practice alone by age to age 20. By age 20, both the best and good violinists had accumulated more than 10 000 h of practice alone on average.

### 3.5.4. Accumulated teacher-designed practice

*Original study analyses.* Ericsson *et al.* [1] did not report estimates of teacher-designed practice independent of 'practice alone'.

*Replication study analyses.* For accumulated teacher-designed practice until age 18, there was again a main effect of group, $\chi_2^2 = 10.74$, $p = 0.005$, $\eta^2 = 0.23$. As with practice alone, the best violinists ($M = 6251$, 95% CI [4293, 8210], range = 2720–14 352) had not accumulated significantly more teacher-designed practice by age 18 than the good violinists ($M = 6821$, 95% CI [4482, 9160], range = 2236–14 612), $t_{24} = -0.37$, $p = 0.717$, $d = -0.15$. The good violinists had accumulated significantly more teacher-designed practice by age 18 than the less accomplished violinists ($M = 2799$, 95% CI [1862, 3735], range = 455–6032), $t_{15.75} = 3.13$, $p = 0.007$, $d = 1.28$. See figure 5.

Estimates of accumulated practice alone until age 18 and accumulated teacher-designed practice until age 18 were highly correlated, $r_{37} = 0.72$, $p < 0.001$. Additionally, the similar effect sizes attained between the two estimates of deliberate practice, $\eta^2 = 0.26$, 95% CI [0.03, 0.44] and $\eta^2 = 0.23$, 95% CI [0.02, 0.41], indicate that the two estimates explain similar amounts of performance variance.

## 4. Discussion

We attempted to replicate the seminal study on deliberate practice theory: Ericsson *et al.*'s [1] study on violin experts. The replication was motivated by several factors. First, this study has had a major impact in both the scientific community and the larger public interested in how to achieve greatness. Part of the reason for this impact is the large effect reported in the original study. A replication of results with improved methods and analyses increases confidence in the original study's surprisingly large findings, whereas a failed replication with improved methods and analyses suggests that the

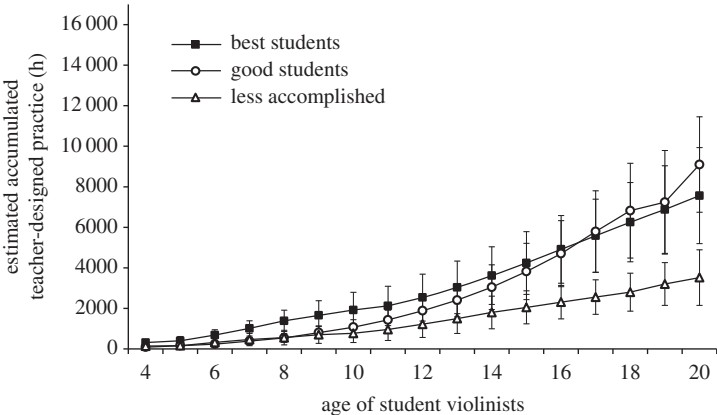

**Figure 5.** Accumulated amount of teacher-designed practice as a function of age for the three skill groups. Error bars represent 95% confidence intervals.

original findings could be due to potential bias or Type I error. Indeed, the original study's methods lent themselves to potential bias that was correctable via double-blind methods, which was another motivation for conducting this replication. Finally, the original study's analyses were conducted in such a way as to make finding statistical significance easier than traditional methods, making it unclear whether the results would replicate with traditional analyses.

In the original study, Ericsson *et al*. [1] found a significant difference between (i) the best and good violinists, and (ii) between the best and good violinists combined as a single group and the less accomplished violinists. They concluded, 'Hence, there is complete correspondence between the skill level of the groups and their average accumulation of practice time alone with the violin' (p. 379). We reproduced Ericsson *et al*.'s [1] methodology but employed a double-blind procedure. We also conducting analyses better suited to the study design.

We did not replicate Ericsson *et al*.'s [1] major result of 'complete correspondence between the skill level of the groups and their average accumulation of practice time alone with the violin' (p. 379). While the less accomplished violinists had accumulated less practice alone than the more accomplished groups, we found no statistically significant differences in accumulated practice alone to age 18 between the best and good violinists. In fact, the *majority* of the best violinists had accumulated *less* practice alone than the average amount of the good violinists. The results were similar when restricting practice estimates to only activities that were designed by a teacher.

Further, the size of the effect did not replicate. Ericsson *et al*.'s [1] comparison of practice alone between the best and good violinists combined as a single group and the less accomplished violinists explained 48% of the variance in performance. Our comparison of practice alone among the three groups explained 26% of the variance, which is similar to 23%, the meta-analytic average amount of performance variance explained by deliberate practice in the music domain [6]. To be clear, explaining 26% of performance variance is not an inconsequential amount. However, this amount does not support the claim that performance levels can '*largely* be accounted for by differential amounts of past and current levels of practice' (p. 392, emphasis added).

Our grouping by skill level via faculty nomination and department was in line with Ericsson *et al*.'s methods. As was the case in Ericsson *et al*. [1], the validity of practice estimates did not differ by group, suggesting that the practice estimates are reliable indicators of actual practice. Why then, might our results have differed from Ericsson *et al*.'s results?

One possibility for the different findings could be differing levels of expertise. While our method of recruitment followed Ericsson *et al*. [1] and our violinists appeared to have the same *relative* difference in skill from each other, they may have an overall higher level of expertise than Ericsson *et al*.'s [1] violinists (e.g. the current violinists had entered many more competitions than those in 1993). If this is the case, it could be that the importance of deliberate practice diminishes at high levels of expertise in music, as has been demonstrated in sports [7].

Our results also might have differed based on the different methods used between the two studies. We conducted a direct replication with exceptions: our study was double-blind rather than non-blinded and we conducted traditional analyses rather than analyses that decreased the critical $F$-statistic needed to find significance.

To this end, Ericsson *et al.*'s [1] results might have been influenced by response bias, experimenter-expectancy bias or a combination of the two. It is unclear how much information Ericsson *et al.* gave to participants, though the original interview materials indicate that participants were told the purpose of the study. When participants are not blind to the purpose of the study or their group membership, they may consciously or subconsciously change their responses to fit the study's goals [10]. Additionally, interview procedures provide ample opportunity for interviewers to subconsciously influence participants [9]. To counter these potential biases, our study was double-blind and all methods and analyses were preregistered.

A final possibility is that Ericsson *et al.*'s [1] results were due to Type I error. Ericsson *et al.*'s [1] method of conducting two ANOVAs per question, and in particular comparing the best and good violinists but using the full sample degrees of freedom, increases the chances of finding $p < 0.05$. We used analyses appropriate for the study design: conducting ANOVAs that included all three groups and planned comparisons, which should not have inflated the chance of Type I errors as the original study's did.

Relatedly, in 2014, Ericsson [20] revealed that the 95% confidence interval around the mean accumulated hours of practice alone to age 18 for the best violinists in Ericsson *et al.* [1] was 2894–11 926 h; note that the mean for the good violinists in Ericsson *et al.* [1] was 5301 h, which falls within this range. It is perhaps surprising that Ericsson *et al.* found significant group differences between the best and good violinists in accumulated practice alone to age 18. Indeed, at least one and probably more of the best violinists from Ericsson *et al.* [1] must have had less accumulated practice than some of the less accomplished good musicians, yet were able to 'catch up' to the best individuals. This result contradicts Ericsson *et al.*'s [1] claim that 'it is impossible for an individual with less accumulated practice at some age to catch up with the best individuals' (p. 393).

## 4.1. Multiple definitions of deliberate practice

To the best of our knowledge, the present study was the first to test and compare both definitions of deliberate practice—practice alone and teacher-designed practice. Indeed, we can find no record of any empirical study of deliberate practice that has operationally defined deliberate practice as only activities designed by a teacher. Ericsson *et al.* [1] are no exception; they do not appear to have measured teacher-designed practice.

Despite this, Ericsson [13,15,16] has sometimes argued that practice activities *need to* be designed by a teacher to qualify as deliberate practice. In the context of arguing against the results of a meta-analysis [6] that found deliberate practice was less important that Ericsson *et al.* [1] claimed, Ericsson (unpublished manuscript, retrieved from: https://psy.fsu.edu/faculty/ericssonk/ericsson.hp.html, page numbers with 'S' refer to Supplemental Materials) argued that many of the included studies should have been excluded because they do not meet the criteria for deliberate practice. Specifically, he (Ericsson, unpublished manuscript) stated, 'The absence of a teacher for all or most of the accumulated practice time violates the definition [of deliberate practice]' (p. S4). He rejected a number of studies included in the meta-analysis, including several of his own studies, because the '[a]rticles do not record a teacher or coach supervising and guiding all or most of the practice' (p. S27).

In contrast with the definition of deliberate practice where activities need to be designed by a teacher, Ericsson [13,15,16] has sometimes argued that practice activities do *not* need be designed by a teacher to qualify as deliberate practice. In line with this definition, Ericsson *et al.* [1] asked participants to estimate hours of 'practice alone' with no apparent restriction to teacher-designed practice. Further, Ericsson *et al.* used the fact that 'practice alone' was rated as most relevant to improvement of violin performance as evidence for the deliberate practice theory. They [1] state: 'In agreement with our theoretical framework, violinists rated practice alone as the most important activity related to improvement of violin performance' (p. 375).

Our study makes a novel contribution by providing evidence about both definitions of deliberate practice. As in Ericsson *et al.* [1], we found that the expert violinists rated practice alone as the most important activity for improving their violin performance. Teacher-designed practice was also rated as highly relevant, but significantly less relevant than practice alone, $p = 0.009$, $d = -0.42$. Accumulated amounts of practice alone (26%) and teacher-designed practice (23%) explained similar amounts of variance in performance.

If deliberate practice is assumed to be the most important activity for improving performance [1], then our results do not support the notion that practice activities *need to* be designed by a teacher to qualify as deliberate practice. That is, the experts in our study did not perceive teacher-designed practice to be the most important type of practice for improving their performance, and the data

support the experts' perception: amount of teacher-designed practice did not predict variance in performance better than amount of practice alone. Amount of teacher-designed practice explained numerically (though not significantly) less variance in performance than practice alone. Simply put, there is no evidence to suggest that teacher-designed practice activities are more relevant to improving performance than practice activities designed by the performer.

Rather, our findings suggest one of two definitions of deliberate practice should be adopted. The first possibility is that deliberate practice (at least in classical music) should clearly and consistently be defined as 'practice alone'. Both Ericsson et al.'s [1] results and the current replication study's results indicate that experts perceive 'practice alone' as most important for improving violin performance.

The other possibility is that deliberate practice should follow Ericsson's [14,17,18] definition that practice activities do *not* need to be designed by a teacher to qualify as deliberate practice. Amount of teacher-designed practice and amount of practice alone were moderately correlated and explained similar amounts of variance in violin performance. Further, Ericsson's [14,17,18] definition that activities can be designed by a teacher *or* the performers themselves is indicative of how deliberate practice is often operationalized in the literature. This includes studies by Ericsson (e.g. [21,22]), and studies in the meta-analysis against which Ericsson (unpublished manuscript) argued.

We believe that theoretical definitions should be empirically tested and not changed depending on the argument. As an example of such a change based on argument, take Tuffiash et al.'s [22] study of expert Scrabble players. Tuffiash et al. [22] described the experts' practice as 'activities that best met the theoretical description of deliberate practice' (p. 131). And, citing that study, Ericsson et al. [23] later described those same activities as 'meeting the criteria of deliberate practice' (p. 9). However, when arguing against the meta-analytic [6] results, Ericsson (unpublished manuscript) rejected this same study because the activities 'violate our original definition of deliberate practice' (p. 4). Definitions of key theoretical terms must be consistent in order to accumulate evidence for or against a theory.

# 5. Conclusion

Using a double-blind procedure and analyses that did not inflate Type I error rates, the main result from Ericsson et al. [1]—that there was complete correspondence between accumulated amount of practice alone and skill level on the violin among elite performers—was not replicated. Our results were similar when examining the role of teacher-designed practice. Our findings suggest that when controlling for biases and Type I error inflation, (i) amount of deliberate practice explains substantially less variance in performance among expert violinists than reported in the original study, and (ii) among more accomplished, elite performers, amount of deliberate practice cannot account for why some individuals acquire higher levels of expert performance than others.

Ethics. This study was approved by the Case Western Reserve University Institutional Review Board. All participants provided informed consent.
Data accessibility. The materials and data are publicly available on the Open Science Framework at https://osf.io/4595q/. The link is included in the manuscript.
Authors' contributions. B.N.M. conceived of the study, wrote the preregistration, coordinated the study, carried out the statistical analyses and drafted the manuscript. M.M. coordinated data collection, collected data and provided revisions. Both authors gave final approval for publication.
Competing interests. We have no competing interests.
Funding. The first author's start-up funds contributed to this study. No external funds contributed to this project. Both authors were employees of Case Western Reserve University during the planning of this study.
Acknowledgements. Thank you to David Gilson at the Cleveland Institute of Music for access to faculty and students and to Victoria F. Sisk for assistance with data collection.

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
