## [Reviewer comments · Royal Society Open Science]

Review History

RSOS-190327.R0 (Original submission)

Review form: Reviewer 1

Do you have any ethical concerns with this paper?

No

Have you any concerns about statistical analyses in this paper?

No

Recommendation?

Accept in principle

Comments to the Author(s)

This looks like a well thought out and important study to conduct given the influence of the original 1993 study. I look forward to finding out the results.

Review form: Reviewer 2

Do you have any ethical concerns with this paper?

No

Have you any concerns about statistical analyses in this paper?

No

Recommendation?

Accept in principle

Comments to the Author(s)

Comments for the Authors:

This study is superbly designed. Indeed, many of the shortcomings of the original study are corrected, while still enabling the authors to conduct a meaningful replication of the original piece of research. With the exception of one possibility, the statistical analyses were all meaningful to me. This is an ambitious design, and I recommend going forward with the study.

The Time 1 assessment protocol is impressive. Very systematic and thoroughly thought out; I especially liked the Swedish Music Discrimination Task and the Raven assessments. Yet, all of the proposed assessments seems highly appropriate to me. The only other thing I might think of is the Wonderlic (Gottfredson, 1997), given the familiarity of this measure to occupational psychology broadly defined. I know that Raven is translatable into IQ units (Jensen et al, 1988), and I very much like that you are using the Raven; but, if possible, the Wonderlic would be highly complementary.

The one statistical analysis that gave me pause was the discriminant function analysis noted on page 37. I have used this application in my own research on multiple occasions, and I have found it to be most informative (as it would be in the current context). But is sample size an issue here given the number of assessments relevant to this analysis? Like factor analysis and multiple regression, the rule of thumb is around 10 to 20 subjects per variable (15 is typically seen as fine). If sample size could be increased to handle this appropriately, I totally agree discriminant function analyses would be a nice feature of this study and would complement the excellent design that the authors have crafted.

Overall, the authors are to be congratulated for detailing and very well-designed study that is important.

References:

Gottfredson, L. G. (1997). Why g matters: The complexity of everyday life. *Intelligence*, 24, 79-132.

Jensen, A. R. et al. (1988). Equating the Standard and Advanced Forms of the Raven Progressive Matrices. *Educational and Psychological Measurement*, 48, 1091-1095.

Decision letter (RSOS-190327.R0)

11-Mar-2019

Dear Dr Macnamara

On behalf of the Editors, I am pleased to inform you that your Manuscript RSOS-190327 entitled "The Role of Deliberate Practice in Expert Performance: Revisiting Ericsson, Krampe, & Tesch-Römer (1993)" deemed suitable for in-principle acceptance in Royal Society Open Science subject to minor revision in accordance with the referee and editor suggestions. Please find their comments at the end of this email.

The reviewers and handling editors have recommended publication, but also suggest some minor revisions to your manuscript. Therefore, I invite you to respond to the comments and revise your manuscript.

Please you submit the revised version of your manuscript within 7 days (i.e. by the 19-Mar-2019). If you do not think you will be able to meet this date please let me know immediately.

When submitting your revised manuscript, you will be able to respond to the comments made by the referees and upload a file "Response to Referees" in the "File Upload" step. You can use this to document any changes you make to the original manuscript. In order to expedite the processing of the revised manuscript, please be as specific as possible in your response to the referees.

Full author guidelines can be found here

<http://rsos.royalsocietypublishing.org/page/replication-studies#AuthorsGuidance>.

Kind regards,
Professor Chris Chambers
Royal Society Open Science
openscience@royalsociety.org

on behalf of Chris Chambers (Registered Reports Editor, Royal Society Open Science)
openscience@royalsociety.org

Associate Editor Comments to Author (Professor Chris Chambers):

Two expert reviewers have now appraised the manuscript. The reviews are positive, with both reviewers judging the Stage 1 Primary Criteria to be met, and both recommending Stage 1 in principle acceptance. Reviewer 2, however, raises two questions about the materials and statistical analyses. If it is not possible to alter these then the response to reviewers (and where appropriate, the manuscript itself) should fully justify these elements of the design. A minor revision is recommended to address these points.

Reviewers' comments to Author:

Reviewer: 1

Comments to the Author(s)

This looks like a well thought out and important study to conduct given the influence of the original 1993 study. I look forward to finding out the results.

Reviewer: 2

Comments to the Author(s)

This study is superbly designed. Indeed, many of the shortcomings of the original study are corrected, while still enabling the authors to conduct a meaningful replication of the original piece of research. With the exception of one possibility, the statistical analyses were all meaningful to me. This is an ambitious design, and I recommend going forward with the study.

The Time 1 assessment protocol is impressive. Very systematic and thoroughly thought out; I especially liked the Swedish Music Discrimination Task and the Raven assessments. Yet, all of the proposed assessments seems highly appropriate to me. The only other thing I might think of is the Wonderlic (Gottfredson, 1997), given the familiarity of this measure to occupational psychology broadly defined. I know that Raven is translatable into IQ units (Jensen et al, 1988), and I very much like that you are using the Raven; but, if possible, the Wonderlic would be highly complementary.

The one statistical analysis that gave me pause was the discriminant function analysis noted on page 37. I have used this application in my own research on multiple occasions, and I have found it to be most informative (as it would be in the current context). But is sample size an issue here given the number of assessments relevant to this analysis? Like factor analysis and multiple regression, the rule of thumb is around 10 to 20 subjects per variable (15 is typically seen as fine). If sample size could be increased to handle this appropriately, I totally agree discriminant function analyses would be a nice feature of this study and would complement the excellent design that the authors have crafted.

Overall, the authors are to be congratulated for detailing and very well-designed study that is important.

References:

Gottfredson, L. G. (1997). Why g matters: The complexity of everyday life. *Intelligence*, 24, 79-132.

Jensen, A. R. et al. (1988). Equating the Standard and Advanced Forms of the Raven Progressive Matrices. *Educational and Psychological Measurement*, 48, 1091-1095.

Author's Response to Decision Letter for (RSOS-190327.R0)

See Appendix A.

Decision letter (RSOS-190327.R1)

21-Mar-2019

Dear Dr Macnamara

On behalf of the Editor, I am pleased to inform you that your Stage 1 Replication RSOS-190327.R1 entitled "The Role of Deliberate Practice in Expert Performance: Revisiting Ericsson, Krampe, & Tesch-Römer (1993)" has been accepted in principle for publication in Royal Society Open Science.

You may now progress to Stage 2 and complete the study as approved.

Please note that you must now register your approved protocol on the Open Science Framework (<https://osf.io/rr>), using the "Submit your approved Registered Report" option and then the "Registered Report Protocol Preregistration" option. Please use the Registered Report option even though your article is being accepted as a Stage 1 Replication. Further into the registration process, in the Journal Title field enter "Royal Society Open Science (Replication article type, Results-Blind track)". Please note that a time-stamped, independent registration of the protocol is mandatory under journal policy, and manuscripts that do not conform to this requirement cannot be considered at Stage 2. The protocol should be registered unchanged from its current approved state. Please include a URL to the protocol in your Stage 2 manuscript, and because you submitted via the Results-Blind track please note in the manuscript that the preregistration was performed after data analysis (e.g. "This article received results-blind in-principle acceptance (IPA) at Royal Society Open Science. Following IPA, the accepted Stage 1 version of the manuscript, not including results and discussion, was preregistered on the OSF (URL). This preregistration was performed after data analysis.")

Following completion of your study, we invite you to resubmit your paper for peer review as a Stage 2 Replication. Please note that your manuscript can still be rejected for publication at Stage 2 if the Editors consider any of the following conditions to be met:

- The Introduction and methods deviated from the approved Stage 1 submission (required).
- The authors' conclusions were not considered justified given the data.

We encourage you to read the complete guidelines for authors concerning Stage 2 submissions at: <http://rsos.royalsocietypublishing.org/page/replication-studies#AuthorsGuidance>. Please especially note the requirements for data sharing and that withdrawing your manuscript will result in publication of a Withdrawn Registration.

Once again, thank you for submitting your manuscript to Royal Society Open Science and I look forward to receiving your Stage 2 submission. If you have any questions at all, please do not hesitate to get in touch. We look forward to hearing from you shortly with the anticipated submission date for your stage two manuscript.

Kind regards,
Professor Chris Chambers
Royal Society Open Science
openscience@royalsociety.org

Author's Response to Decision Letter for (RSOS-190327.R1)

See Appendix B.

RSOS-190327.R2 (Revision)

Review form: Reviewer 1

Do you have any ethical concerns with this paper?

No

Have you any concerns about statistical analyses in this paper?

No

Recommendation?

Accept with minor revision

Comments to the Author(s)

Thanks for this paper, it is an important contribution to the literature. I only have a few more suggestions to strengthen the manuscript.

1. You motivate your article in the introduction by explaining how this particular paper has had a large impact both within the academic community and also outside it among the public. Perhaps this could be mentioned as an additional motivating factor for this attempted replication in the beginning of your discussion section given I think this is a key reason why you are attempting to see if the paper stands even with your additional improved method and analysis additions.
2. Though I think you are reasonably fair to Ericsson in regards to his definitional shifts or creeps, at the same time I'd encourage you to reread especially the abstract, intro, and discussion in particular to ensure it is as neutral as possible.
3. You note that the size of the core finding was much smaller. Is this smaller finding still of importance? Is there a way to communicate that clearly to the reader in the abstract and in the discussion/conclusion?
4. It seems that you are basically stating that you tried to replicate this really influential expertise paper and can't really replicate it and in fact the conclusions of the original study are not supported as stated by the original authors. It seems most plausible that the lack of replication is due to the your additional improvements in the methods. Though this is not a direct replication given you've made some improvements. Could perhaps discuss a bit about what this type of replication is? (is it a constructive replication, following Lykken's formulation?)

Thanks for conducting this important research and I hope these suggestions are helpful.

Review form: Reviewer 2

Do you have any ethical concerns with this paper?

No

Have you any concerns about statistical analyses in this paper?

No

Recommendation?

Accept as is

Comments to the Author(s)

Comments for the Authors:

I have read this study twice now and believe that it is an excellent contribution to the literature. The authors have provided a compelling conceptual and statistical rationale for their approach and, in many ways, the way in which they have operationalized each analytic move is exemplary. I was particularly impressed by the way in which they linked each analysis to embedded text (and quotes) that they pulled from the literature.

In addition to executing a superb replication, the authors have added important (additional) controls of their own to advance this area. This is very careful, impressive, and important work.

A few things to think about in crafting a revision:

Abstract

Rather than “operational definition” how about the way in which Ericsson et al. (1993) operationalize deliberate practice (practice alone), and their theoretical but previously unmeasured definition of deliberate practice (teacher-designed practice), ...” This way, your description won’t be tied to earlier ideas about operational definitions (viz., a concept IS what it measures).

Page 3 (first paragraph). I couldn’t help thinking of Epstein’s (2011) excellent book, *The sports gene*.

Because in addition to height and body size, he has several other physical and physiological parameters of human individuality that clearly come into play.

Page 3 (second paragraph). Was it Gladwell’s 10,000-hour rule? I thought this was suggested by someone else earlier.

Pages 13-14 constitute an excellent discussion of “sample size,” very compelling.

Reference

Epstein, D. (2011). *The sports gene: Inside the science of extraordinary athletic performance*. New York, NY: Current.

Decision letter (RSOS-190327.R2)

08-Jul-2019

Dear Dr Macnamara

On behalf of the Editor, I am pleased to inform you that your Stage 2 Replication submission RSOS-190327.R2 entitled "The Role of Deliberate Practice in Expert Performance: Revisiting Ericsson, Krampe, & Tesch-Römer (1993)" has been accepted for publication in Royal Society Open Science subject to minor revision in accordance with the referee suggestions. Please find the referees' comments at the end of this email.

The reviewers and Subject Editor have recommended publication, but also suggest some minor revisions to your manuscript. Therefore, I invite you to respond to the comments and revise your manuscript.

Please also ensure that all the below editorial sections are included where appropriate (a non-exhaustive example is included in an attachment):

- Ethics statement

- Data accessibility

<http://datadryad.org/submit?journalID=RSOS&manu=RSOS-190327.R2>

- Competing interests

- Authors' contributions

- Acknowledgements

- Funding statement

Because the schedule for publication is very tight, it is a condition of publication that you submit the revised version of your manuscript within 7 days (i.e. by the 16-Jul-2019). If you do not think you will be able to meet this date please let me know immediately.

- 1) A text file of the manuscript (tex, txt, rtf, docx or doc), references, tables (including captions) and figure captions. Do not upload a PDF as your "Main Document".
- 2) A separate electronic file of each figure (EPS or print-quality PDF preferred (either format should be produced directly from original creation package), or original software format)
- 3) Included a 100 word media summary of your paper when requested at submission. Please ensure you have entered correct contact details (email, institution and telephone) in your user account
- 4) Included the raw data to support the claims made in your paper. You can either include your data as electronic supplementary material or upload to a repository and include the relevant DOI within your manuscript
- 5) Included your supplementary files in a format you are happy with (no line numbers, Vancouver referencing, track changes removed etc) as these files will NOT be edited in production

Kind regards,
Professor Chris Chambers

Registered Reports Editor
Royal Society Open Science
openscience@royalsociety.org

Editor Comments to Author (Professor Chris Chambers):

The Stage 2 manuscript was returned to the two expert reviewers who assessed it at Stage 1. Both are positive about the submission while offering some suggestions for minor revision. In revising the manuscript, please be sure to make no unnecessary changes to the Introduction and Methods that were assessed and approved at Stage 1. Provided the authors respond thoroughly to reviewers' comments, full acceptance should be forthcoming without requiring further in-depth review.

Reviewers' comments to Author:

Reviewer: 1

Comments to the Author(s)

Thanks for this paper, it is an important contribution to the literature. I only have a few more suggestions to strengthen the manuscript.

1. You motivate your article in the introduction by explaining how this particular paper has had a large impact both within the academic community and also outside it among the public. Perhaps this could be mentioned as an additional motivating factor for this attempted replication in the beginning of your discussion section given I think this is a key reason why you are attempting to see if the paper stands even with your additional improved method and analysis additions.
2. Though I think you are reasonably fair to Ericsson in regards to his definitional shifts or creeps, at the same time I'd encourage you to reread especially the abstract, intro, and discussion in particular to ensure it is as neutral as possible.
3. You note that the size of the core finding was much smaller. Is this smaller finding still of importance? Is there a way to communicate that clearly to the reader in the abstract and in the discussion/conclusion?
4. It seems that you are basically stating that you tried to replicate this really influential expertise paper and can't really replicate it and in fact the conclusions of the original study are not supported as stated by the original authors. It seems most plausible that the lack of replication is due to the your additional improvements in the methods. Though this is not a direct replication given you've made some improvements. Could perhaps discuss a bit about what this type of replication is? (is it a constructive replication, following Lykken's formulation?)

Thanks for conducting this important research and I hope these suggestions are helpful.

Reviewer: 2

Comments for the Authors:

I have read this study twice now and believe that it is an excellent contribution to the literature. The authors have provided a compelling conceptual and statistical rationale for their approach and, in many ways, the way in which they have operationalized each analytic move is exemplary.

I was particularly impressed by the way in which they linked each analysis to embedded text (and quotes) that they pulled from the literature.

In addition to executing a superb replication, the authors have added important (additional) controls of their own to advance this area. This is very careful, impressive, and important work.

A few things to think about in crafting a revision:

Abstract

Rather than “operational definition” how about the way in which Ericsson et al. (1993) operationalize deliberate practice (practice alone), and their theoretical but previously unmeasured definition of deliberate practice (teacher-designed practice), ...” This way, your description won’t be tied to earlier ideas about operational definitions (viz., a concept IS what it measure IS).

Page 3 (first paragraph). I couldn’t help thinking of Epstein’s (2011) excellent book, *The sports gene*.

Because in addition to height and body size, he has several other physical and physiological parameters of human individuality that clearly come into play.

Page 3 (second paragraph). Was it Gladwell’s 10,000-hour rule? I thought this was suggested by someone else earlier.

Pages 13-14 constitute an excellent discussion of “sample size,” very compelling.

Reference

Epstein, D. (2011). *The sports gene: Inside the science of extraordinary athletic performance*. New York, NY: Current.

Author's Response to Decision Letter for (RSOS-190327.R2)

See Appendix C.

Decision letter (RSOS-190327.R3)

23-Jul-2019

Dear Dr Macnamara:

It is a pleasure to accept your Stage 2 Replication entitled "The Role of Deliberate Practice in Expert Performance: Revisiting Ericsson, Krampe, & Tesch-Römer (1993)" in its current form for publication in Royal Society Open Science.

on behalf of Chris Chambers (Subject Editor)
Registered Reports Editor
Royal Society Open Science
openscience@royalsociety.org

Appendix A

Associate Editor Comments to Author (Professor Chris Chambers):

Two expert reviewers have now appraised the manuscript. The reviews are positive, with both reviewers judging the Stage 1 Primary Criteria to be met, and both recommending Stage 1 in principle acceptance. Reviewer 2, however, raises two questions about the materials and statistical analyses. If it is not possible to alter these then the response to reviewers (and where appropriate, the manuscript itself) should fully justify these elements of the design. A minor revision is recommended to address these points.

Reply: Thank you for the opportunity to submit minor revisions.

Reviewers' comments to Author:

Reviewer: 1

Comments to the Author(s)

This looks like a well thought out and important study to conduct given the influence of the original 1993 study. I look forward to finding out the results.

Reply: Thank you for your comments. We look forward to presenting the results.

Reviewer: 2

Comments to the Author(s)

This study is superbly designed. Indeed, many of the shortcomings of the original study are corrected, while still enabling the authors to conduct a meaningful replication of the original piece of research. With the exception of one possibility, the statistical analyses were all meaningful to me. This is an ambitious design, and I recommend going forward with the study.

Reply: Thank you for these comments.

The Time 1 assessment protocol is impressive. Very systematic and thoroughly thought out; I especially liked the Swedish Music Discrimination Task and the Raven assessments. Yet, all of the proposed assessments seems highly appropriate to me. The only other thing I might think of is the Wonderlic (Gottfredson, 1997), given the familiarity of this measure to occupational psychology broadly defined. I know that Raven is translatable into IQ units (Jensen et al, 1988), and I very much like that you are using the Raven; but, if possible, the Wonderlic would be highly complementary.

Reply: Thank you for the suggestion. Ideally, with enough resources, other tests could be included. There are a number of reasons why we don't think the Wonderlic is ideal. First, while it is a popular measure, evidence of its validity is limited. Notably, the Wonderlic does not predict academic performance (Chamorro-Premuzic & Furnham, 2008; Furnham, Chamorro-Premuzic, & McDougall, 2002; McKelvie, 1994) and is inconsistent as a predictor of job performance (Barrick, Mount, & Strauss, 1993; Frei & McDaniel, 1998; Hogan & Hogan, 1995; Lyons, Hoffman, & Michel, 2009; Rode, Arthaud-Day, Mooney, Near, & Baldwin, 2008). Further, recent work by Hicks, Harrison and Engle (2015) found that, when controlling for working memory capacity, the Wonderlic has no direct association with fluid intelligence. Hicks et al. (2015) also found that the Wonderlic varies in whether it predicts working memory capacity or not depending on participants' fluid intelligence. We have more succinctly added this rationale to the manuscript. The Fluid Intelligence measure sub-section now reads:

Fluid intelligence. We included two measures of fluid intelligence: Raven's Advanced Progressive Matrices ^[S5] and Letter Sets ^[S6]. The mean proportion correct on the two measures serves as the composite fluid intelligence score. A number of other intelligence measures could have been included, such as the Wonderlic test ^[S7]. We did not include the

Wonderlic because evidence for its validity is limited both in terms of being a consistent predictor of job performance and in terms of correlations with fluid intelligence (see, e.g., [S8]).

The one statistical analysis that gave me pause was the discriminant function analysis noted on page 37. I have used this application in my own research on multiple occasions, and I have found it to be most informative (as it would be in the current context). But is sample size an issue here given the number of assessments relevant to this analysis? Like factor analysis and multiple regression, the rule of thumb is around 10 to 20 subjects per variable (15 is typically seen as fine). If sample size could be increased to handle this appropriately, I totally agree discriminant function analyses would be a nice feature of this study and would complement the excellent design that the authors have crafted.

Reply: We have removed discriminant function analysis from the analysis plan.

Overall, the authors are to be congratulated for detailing and very well-designed study that is important.

Reply: Thank you for these kind words.

Other changes: We fixed the in-text citations and references of what will be in the supplemental materials to match that of the main document (i.e., numbered).

Appendix B

June 14th, 2019

Dear Prof. Christopher Chambers:

Following our Stage 1 IPA, we are submitting our Stage 2 manuscript.

Thank you for your time.

Sincerely yours,

Brooke N. Macnamara, Case Western Reserve University
Megha Maitra, Case Western Reserve University

Corresponding author:
Brooke N. Macnamara
Department of Psychological Sciences
Case Western Reserve University
Phone: 1+216-368-2681
Email: brooke.macnamara@case.edu

Appendix C

July 12th, 2019

Dear Prof. Christopher Chambers:

Thank you for the opportunity to revise and resubmit. We have responded to all reviewers' comments. See below.

Thank you for your time.

Sincerely yours,

Brooke N. Macnamara, Case Western Reserve University

Megha Maitra, Case Western Reserve University

Corresponding author:

Brooke N. Macnamara

Department of Psychological Sciences

Case Western Reserve University

Phone: 1+216-368-2681

Email: brooke.macnamara@case.edu

Reviewer: 1

1. You motivate your article in the introduction by explaining how this particular paper has had a large impact both within the academic community and also outside it among the public. Perhaps this could be mentioned as an additional motivating factor for this attempted replication in the beginning of your discussion section given I think this is a key reason why you are attempting to see if the paper stands even with your additional improved method and analysis additions.

Reply: This paragraph has been edited to include the major impact of this study as a motivation. The first paragraph in the Discussion now reads as follows:

“We attempted to replicate the seminal study on deliberate practice theory: Ericsson, Krampe, and Tesch-Römer’s^[1] study on violin experts. The replication was motivated by several factors. First, this study has had a major impact in both the scientific community and the larger public interested in how to achieve greatness. Part of the reason for this impact is the large effect reported in the original study. A replication of results with improved methods and analyses increases confidence in the original study’s surprisingly large findings, whereas a failed replication with improved methods and analyses suggests that the original findings could be due to potential bias or Type I error. Indeed, the original study’s methods lent themselves to potential bias that was correctable via double-blind methods, which has another motivation for conducting this replication. Finally, the original study’s analyses were conducted in such a way as to make finding statistical significance easier than traditional methods, making it unclear whether the results would replicate with traditional analyses.”

2. Though I think you are reasonably fair to Ericsson in regards to his definitional shifts

or creeps, at the same time I'd encourage you to reread especially the abstract, intro, and discussion in particular to ensure it is as neutral as possible.

Reply: We have reread the manuscript to ensure we are as neutral as possible.

3. You note that the size of the core finding was much smaller. Is this smaller finding still of importance? Is there a way to communicate that clearly to the reader in the abstract and in the discussion/conclusion?

Reply: We have communicated in the abstract and Discussion that the amount explained is not inconsequential. The abstract now includes this sentence: "Overall, the size of the effect was substantial, but considerably smaller than the original study's effect size." The Discussion now includes this sentence: "To be clear, explaining 26% of performance variance is not an inconsequential amount. However, this amount does not support the claim that performance levels can "*largely* be accounted for by differential amounts of past and current levels of practice" (p. 392, emphasis added)."

4. It seems that you are basically stating that you tried to replicate this really influential expertise paper and can't really replicate it and in fact the conclusions of the original study are not supported as stated by the original authors. It seems most plausible that the lack of replication is due to the your additional improvements in the methods. Though this is not a direct replication given you've made some improvements. Could perhaps discuss a bit about what this type of replication is? (is it a constructive replication, following Lykken's formulation?)

Reply: We clarify in the Discussion that we conducted a direct replication with exceptions. This seems to be the most apt description. That is, it is not a constructive replication, where the replicating authors choose any methods they wish (i.e., a conceptual replication), because, this replication followed the methods from the original study as directly as possible with the exception of fixing what we perceive to be as methodological and analytical flaws. We restructured the Discussion to follow this reasoning and discuss that the results differences might have been due to methods differences. Specifically, after asking why our results might have differed, we moved the paragraph about expert level differences up first, then added: "Our results also might have differed based on the different methods used between the two studies. We conducted a direct replication with exceptions: Our study was double-blind rather than non-blinded and we conducted traditional analyses rather than analyses that decreased the critical F -statistic needed to find significance.

To this end..."

Thanks for conducting this important research and I hope these suggestions are helpful.

Reply: Thank you for your feedback.

Reviewer: 2

Comments for the Authors:

I have read this study twice now and believe that it is an excellent contribution to the literature. The authors have provided a compelling conceptual and statistical rationale for

their approach and, in many ways, the way in which they have operationalized each analytic move is exemplary. I was particularly impressed by the way in which they linked each analysis to embedded text (and quotes) that they pulled from the literature.

In addition to executing a superb replication, the authors have added important (additional) controls of their own to advance this area. This is very careful, impressive, and important work.

Reply: Thank you for these kind words.

A few things to think about in crafting a revision:

Abstract

Rather than “operational definition” how about the way in which Ericsson et al. (1993) operationalize deliberate practice (practice alone), and their theoretical but previously unmeasured definition of deliberate practice (teacher-designed practice), ...” This way, your description won’t be tied to earlier ideas about operational definitions (viz., a concept IS what its measure IS).

Reply: We have made this change. It now reads: “we examined the way Ericsson et al. (1993) operationalized deliberate practice (practice alone), and their theoretical but previously unmeasured definition of deliberate practice (teacher-designed practice), and compared them.”

Page 3 (first paragraph). I couldn’t help thinking of Epstein’s (2011) excellent book, *The sports gene*.

Because in addition to height and body size, he has several other physical and physiological parameters of human individuality that clearly come into play.

Reply: We agree. It is an excellent book and he described a number of factors important for sports. We are concerned with discussing Epstein’s book in the first paragraph though given that the current study is not on sports. We believe this would be tangential and break up the flow.

Page 3 (second paragraph). Was it Gladwell’s 10,000-hour rule? I thought this was suggested by someone else earlier.

Reply: We have changed “his” to “the.”

Pages 13-14 constitute an excellent discussion of “sample size,” very compelling.

Reply: Thank you.

Reference

Epstein, D. (2011). *The sports gene: Inside the science of extraordinary athletic performance*. New York, NY: Current.